

# Impact of differential stress on fracture due to volume increasing hydration

Jeremiah J. McElwee[1], Ikuko Wada[1], Kazuki Yoshida[2], Hiroyuki Shimizu[3], Atsushi Okamoto[4]

[1]Department of Earth and Environmental Sciences, University Minnesota Twin Cities, Minneapolis, 55406, USA
[2]Institute of Materials Structure Science, High Energy Accelerator Research Organization, Tsukuba, 305-0801, Japan
[3]Geotechnical Analysis Group, Advanced Analysis Department, Civil Engineering Design Division, Kajima Corporation, Tokyo, Japan
[4]Graduate School of Environmental Studies, Tohoku University, Miyagi, 980-8578, Japan

*Correspondence to:* Jeremiah J. McElwee (mcelw020@umn.edu)

**Abstract.** The volume increase that accompanies many hydration reactions can stress and fracture the surrounding rock, a process commonly called reaction-induced fracture. Reaction-induced fracture accelerates the rate of hydration by creating new pathways for fluids to migrate into reactive rock and by generating new reactive surface areas. The evolution of reaction-induced fracture also depends on the background stress state, which varies among different tectonic environments. We investigate the impact of tectonic stresses on reaction-induced fracture, using 2-D hydraulic-chemical-mechanical distinct element models. The results indicate that the general pattern of reaction-induced fracture depends on the orientation of background tectonic stresses relative to fluid-supplying channels. A spalling fracture pattern characterized by short cracks parallel to and along fluid-supplying channels occurs when the maximum principal tectonic stress is parallel to the channels whereas a branching fracture pattern characterized by long tensile cracks propagate in a hierarchical manner into unreacted part of the rock is expected when the tectonic stress is hydrostatic or when the maximum principal tectonic stress is normal to fluid-supplying channels. Spalling localizes hydration and fluid flow along the channels whereas branching promotes spatially extensive hydration and fluid flow away from the fluid supply. The results indicate tectonic stresses may guide the hydration distribution in the oceanic lithosphere at mid-ocean ridges and outer rises and in the cold mantle wedge corner in subduction zones.

## 1 Introduction

Hydration of Earth materials is frequently accompanied by a solid volume increase and reaction-induced stress (also called crystallization pressure/stress) that can cause fractures in the surrounding rock, impacting subsequent fluid flow and hydration (*Macdonald and Fyfe; 1985; Uno et al., 2022; Okamoto and Shimizu, 2015; Yoshida et al., 2020; Renard 2021; Zheng et al., 2018; Plümper et al., 2012, 2022; Rudge et al., 2010; Jamtveit et al., 2009; Jamtveit et al., 2008; Keleman and Hirth, 2012; Malvoisin et al., 2017; Evans et al., 2020*). For example, hydration of ultramafic mantle rock to serpentinites results in up to ~50% volume increase (*Klein and Le Roux, 2020; Malvoisin et al., 2020; Coleman and Keith; 1971*), and reaction-induced fractures have been inferred from serpentinites from mid-ocean ridges (*Shimizu and Okamoto, 2016; Roumejon and Cannat, 2014; MacDonald and Fyfe, 1985*), mantle wedge corners in subduction zones (*Dandar et al., 2019; Uno and Kirby, 2019*), and near-surface environments (*Malvoisin et al., 2017; Katayama et al., 2021*). Such reaction-induced fractures likely play a role in local fluid flow. Therefore, understanding reaction-induced fracture during hydration reactions is potentially critical to understanding fluid migration and the distribution of hydration.



In addition to the reaction-induced stress, tectonic stresses impact the formation, distribution, and orientation of fractures (e.g., *Seno, 2005*). At mid-ocean ridges and outer rises along subduction margins, the state of stress in the upper portion of the lithosphere is approximately deviatoric tension parallel to the direction of plate motion due to gravitational sliding and plate bending, respectively, as inferred from the formation of normal faults (e.g., *Ranero et al., 2003; Buck et al., 2005*). In subduction

zones, the tectonic stresses in the arc-ward part of the forearc region (the inner forearc) of the overriding lithosphere vary from margin-normal deviatoric tension to compression (*Wang and He, 1999; Wang and Suyehiro, 1999; Balfour et al., 2011; Yoshida et al., 2015; Saito et al., 2018*). Experiments have been conducted on volume-expanding hydration reactions under axial compression, investigating reaction-induced fractures (*Zheng et al., 2018; Okamoto et al., 2025*). However, how the fracture behavior changes with differential stress is still unclear.


Here, we study the impact of reaction-induced stress and background differential stress on fracture evolution through 2-D hydraulic-chemical-mechanical modeling (*Shimizu et al., 2011; Okamoto and Shimizu, 2015*). Previous studies that employ similar numerical modeling approaches investigate reaction-induced fracture at hydrostatic conditions (*Yoshida et al., 2020; Shimizu and Okamoto, 2016; Zheng et al., 2019; Malthe-Sørenssen et al., 2006; Jamtveit et al., 2009*). They report that the solid

volume change associated with hydration leads to fracture that enhances the permeability of the reacting rock and its surrounding. However, the evolution of reaction-induced fracture is likely impacted by the stresses associated with a given tectonic environment. Here, we take a step further by applying a range of differential stresses and investigate the relation between the differential stress and fracture evolution.

## 2 Methods

We use the 2-D numerical code DEFraG previously developed by *Shimizu et al. (2011)* and modified for simulating reaction-induced fracture by *Okamoto and Shimizu (2015).* The code employs the distinct element method (DEM; *Cundall and Strack, 1979; Potyondy and Cundall, 2004*) and calculates the volume expansion in response to increasing fluid pressure, stress due to the volume expansion, and fracture due to the induced stress. A previous version of this code and similar codes have been used to investigate reaction-induced fracture at hydrostatic conditions (*Yoshida et al., 2020; Shimizu and Okamoto, 2016*). These studies

provide a foundation for the work presented here and support the veracity of the approach. The modeling approach is described in detail in *Shimizu et al. (2011)* and *Okamoto and Shimizu (2015)*. Here, we provide a brief description.





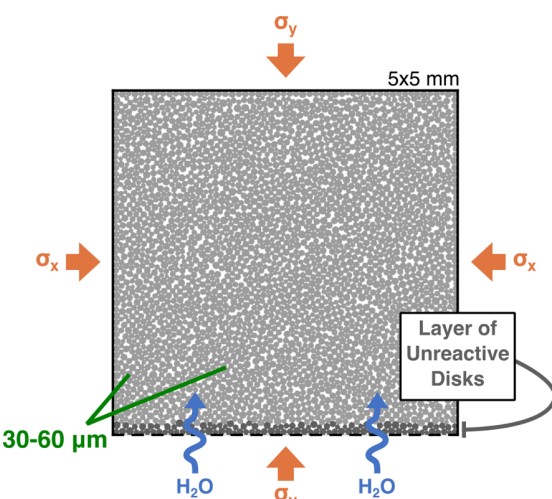

**Figure 1. Model setup. Fluid pressure of 30 MPa is prescribed at the base of the model (black dashed line). The left, right, and top**
**black solid lines indicate no-flow boundary conditions. Disk size is not necessarily representative of mineral grain size, but rather the**
**length scale of heterogeneity in the rock strength. A highly permeable layer that consists of unreactive disks is placed at the base of the**
**model to minimize boundary effects on the bottom-most reactive disks.**

### 2.1 Mechanical Modeling Approach

The modeled system is represented by a lattice of disks connected by elastic bonds, representing a small portion of rock relative
to the length scale of the assumed fluid source (**Fig. 1**). The elastic properties of the bonds are described in terms of shear,
tensile, and rotational stiffnesses. The tensile and rotational stiffnesses are calculated based on Bernoulli-Euler beam theory, and
the shear stiffness is calculated based on a chosen ratio between shear and tensile stiffnesses. At critical stresses the bonds break
and are replaced by cracks (Hereafter we refer to breakage between two disks as a crack and longer breakage as a fracture).

Because the model is formulated as a large number of bonds and disks, the macroscopic behavior is emergent such that
parameters of the bonds must be calibrated to produce a desired macroscopic behavior. The calibration procedure is detailed in
*Shimizu and Okamoto (2016)*, and we use their calibrated microscopic mechanical properties: constant values of 30 MPa, 116
MPa, and infinity for the normal, shear, and rotational strengths, respectively, and 348 GPa and 126 GPa for Young's modulus of
the bonds between unreacted and fully reacted disks, respectively. In addition, we prescribe the ratios of shear to normal stiffness

and rotational to normal stiffness to be 0.241 and 1, respectively, for unreacted disks, and 0.206 and 1, respectively, for fully
reacted disks. Although we vary the elastic properties with the degree of hydration, previous modeling studies that employ
similar approaches have reported that variations in both elastic properties and strengths of the rocks have relatively small effects
on the overall pattern of reaction-induced fracture under hydrostatic conditions (*Zhang et al., 2019; Okamoto and Shimizu,*
*2015*).


To verify that the chosen microscopic parameters produce reasonable macroscopic behavior, uniaxial compression and uniaxial
tension strength tests were simulated using models composed completely of unreacted or fully reacted disks (**Fig. S1**). These
microscopic parameter values yield macroscopic elastic properties (i.e., Young's modulus and Poisson's ratio) of unreacted and
reacted disks that are comparable to values reported in the literature for peridotite and an average of those for pure lizardite and





pure brucite, respectively (*Christensen, 2004*) (**Table 1**). These analogs are useful given the prevalence of peridotite hydration to serpentinite in ultramafic tectonic environments. Additionally, both unreacted and reacted material exhibits approximately 10 MPa uniaxial tensile strength and 100 MPa uniaxial compressive strength, which are comparable to the strengths common rock (*Pollard and Fletcher, 2005*). The choice of these values maintains consistency with previous studies (*Shimizu and Okamoto, 2016*) and is also useful for comparison.


All models are 5 mm in width and height and consist of 3,151 disks with 30–60 $\mu$m radii (average of 44.7 $\mu$m). The confining pressure $\sigma_c$ is 1 MPa for all models. For non-hydrostatic conditions, we vary the magnitudes of the horizontal and vertical stresses ($\sigma_x$ and $\sigma_y$, respectively) (**Fig. 1**) and describe the stress condition as either horizontal deviatoric compression or tension (hereafter simply referred to as compression and tension, respectively) (**Table 2**). The left, right, and top boundaries are

rigid but can move to maintain the applied stress.

**Table 1: Macroscopic mechanical properties of models composed entirely of unreacted and reacted disks.**

| Parameter | Unreacted | Fully reacted |
|---|---|---|
| Uniaxial Compression Strength (MPa) | 98.2 | 100.9 |
| Uniaxial Tensile Strength (MPa) | 10.3 | 9.9 |
| Young's Modulus (GPa) | 162 | 55.0 |
| Poisson's Ratio | 0.22 | 0.23 |

**Table 2: Stresses applied to the boundaries of the model**

| Model ID | $\sigma_x$ (MPa) | $\sigma_y$ (MPa) | $\sigma_x - \sigma_y$ ($\sigma_{diff}$; MPa) | |
|---|---|---|---|---|
| H1 | 1 | 1 | 0 | Hydrostatic |
| T1 | 1 | 6 | -5 | Horizontal deviatoric tension |
| T2 | 1 | 11 | -10 | |
| C1 | 6 | 1 | 5 | Horizontal deviatoric compression |
| C2 | 11 | 1 | 10 | |


## 2.2 Hydraulic Modeling Approach

Pore space between disks is occupied by fluids. Due to the circular shape of the disks, the porosity in the model is unrealistically large. Therefore, we scale the pore space in the simulation to be 1% of the actual pore space.



The initial fluid pressure in all models is set everywhere to the minimum value of fluid ($P_{min}$) that needs to be overcome before the reaction can proceed. The maximum fluid pressure ($P_{max}$) at which the reaction proceeds at a prescribed maximum rate ($Z_{max}$) is maintained at the base of the model. The other three boundaries are impermeable. We use $P_{min}$ of 29 MPa and $P_{max}$ of 30 MPa. Pore fluids do not exert pressure on the model boundaries. Therefore, the simulation behavior is not sensitive to the magnitude of $P_{max}$. Additionally, the difference between $P_{max}$ and $P_{min}$ is less than the strength of the bonds such that

hydrofracturing does not occur. This choice is made to evaluate the effects of reaction-induced and tectonic stresses on fracture in the absence of hydrofracturing. Fluids may exert significant stresses and contribute to fracture in tectonic settings where the pore pressure is high and sharp fluid pressure gradients are present. The scenario modeled in this study is one where pore pressure gradients are relatively low such that hydrofracture is negligible. The relation between the reaction rate and the fluid pressure is further described in **Sect. 2.3** below.


    Fluid flow driven by fluid pressure gradients occurs between pores via channels between bonded disks and cracks that form between disks. We refer to these two types of flow as matrix flow and fracture flow, respectively. Matrix flow represents flow governed by the initial permeability of the reacting material governed by, for example, connectivity of pores, diffusion, or thermal cracking (e.g., *Yoshida et al., 2020; Roumejon and Cannat, 2014; Boudier et al., 2009*). In both cases, imaginary pipes

are placed between disks to represent flow channels, and the volumetric flow ($Q$) through the channel is calculated using the laminar flow equation

$$Q = \frac{w^3}{12\nu L} \Delta P \qquad (1)$$

where $\Delta P$ is the pressure gradient, $\nu$ is the fluid viscosity, $L$ is the length of the channel, and $w$ is the aperture of the channel. For two disks whose bond is broken but are in contact, the channel aperture depends on the force ($F$) acting on the channel

$$w = w_0 \frac{F_0}{F + F_0} \qquad (2)$$

where $w_0$ is the predefined aperture of a channel subject to no forces and $F_0$ is the force necessary to close the channel halfway (i.e. $F_0 = \frac{1}{2} k_n w_0$; where $k_n$ is the elastic stiffness of the contact). For two disks that have a broken bond and are not in contact, the aperture is taken as the sum of the disk separation distance and $w_0$.

## 2.3 Chemical Modelling Approach

When disks are in contact with fluid and the fluid pressure is greater than $P_{min}$, they react and consume the fluid. In our simulation, the complete reaction is accompanied by a 50% solid volume increase and a 20% total (solid + fluid) volume decrease. These values are comparable to those reported for serpentinization, which involves dissolution and replacement of the reacting material (*Malvoisin et al., 2015; Klein and Le Roux, 2020; Malvoisin et al., 2021; Malvoisin et al., 2020*). In the model, this volume expansion occurs as a radially symmetric expansion of the reacting disks. The elastic parameters of partially reacted

disks vary linearly between those of unreacted and completely reacted disks (**Sect. 2.1**) as the reaction of each disk progresses.

    Disks consume fluids at a reaction rate $Z$. We use the following reaction rate equation (*Shimizu and Okamoto, 2015*):

$$Z = Z_{max} \qquad\qquad if\ P \geq P_{max} \qquad (3)$$

$$Z = Z_{max}\left(\frac{P - P_{min}}{P_{max} - P_{min}}\right) \quad if\ P_{min} < P < P_{max} \qquad (4)$$



$Z = 0$             $if\ P\ \leq P_{min}$     (5)

where $P$ is the local fluid pressure. A positive, linear, fluid-pressure dependence is reported for serpentinization (*Malvoisin et al., 2015; Wegner and Ernst 1983*). In the model, if $Z_{max}$ is too high, large stress gradients develop in a single timestep, resulting in artificial spontaneous fracture, and the resulting cracks have previously been referred to as "inertial cracks" (*Shimizu and Okamoto, 2016*). Previous studies report stable reaction rates that minimize the occurrence of inertial cracks at hydrostatic stress

conditions (e.g., *Shimizu and Okamoto, 2016; Okamoto et al., 2017*) but not under differential stresses. Through a series of simulations with different $Z_{max}$ values, we find that the crack density loses its reaction-rate dependence when $Z_{max}$ is lower than 25 s$^{-1}$, and inertial cracks are largely absent although the orientation of cracks and the ratio of shear to tensile cracks vary with the reaction rate, impacting the resulting fracture texture (**Fig. S2**). Further analyses indicate that the ratio of shear to tensile cracks converges near $Z_{max} = 5$ s$^{-1}$. To avoid the formation of artificial cracks and minimize the impact of $Z_{max}$ on fracture

texture, we use $Z_{max} = 5$ s$^{-1}$ for all models except the hydrostatic case, for which we use $Z_{max} = 1$ s$^{-1}$ to reduce the fluid flow rate to avoid anomalously long computation times (**Text S1**).

In the model calculation, very small time steps ($\sim2.5 \times 10^{-10}$ s) are required to maintain numerical stability. To keep computation times reasonable, reaction rates much faster than expected to be in nature are used, and fluid flow rates are scaled by the reaction

rate:

$$\psi_m = \frac{Characteristic\ matrix\ flow\ rate}{Characteristic\ reaction\ rate} = \frac{Q_m}{Q_r} \quad (6)$$

$$\psi_f = \frac{Characteristic\ fracture\ flow\ rate}{Characteristic\ reaction\ rate} = \frac{Q_f}{Q_r} \quad (7)$$

where $Q_m$ is the characteristic volumetric matrix flow rate between two bonded disks, $Q_f$ is the characteristic volumetric fracture flow rate in a crack, and $Q_r$ is the characteristic volumetric rate of fluid consumption, as described in *Shimizu and Okamoto*

*(2016)*. $Q_m$ and $Q_f$ are calculated based on chosen values for $w_0$ for channels between bonded disks and cracks, respectively. Laboratory experiments on the hydration of periclase to brucite, commonly used as a proxy for general volume increasing reactions, and computer modeling work on reaction-induced fracture indicate that the behavior of these systems depends on $\psi$ rather than the absolute reaction rate and permeability (*Uno et al., 2022; Shimizu and Okamoto, 2016, Ulven et al., 2014b*). In particular, reaction-induced fracture only occurs when $\psi_m$ is below a critical value of 100, and textures similar to those seen in

nature develop only when $\psi_f$ is above a critical value of 1000 (*Uno et al., 2022; Shimizu and Okamoto, 2016*). In this study, we run models with either moderate or high value of $\psi_m$ (1 or 10) for reactive disks and use $\psi_f$ of 10,000 for all models.

In all cases, fluid is supplied at the base of the model. We include a layer of unreactive material at the base of the model domain such that the boundary effects on the reactive discs are minimized (**Fig. 1**). We use $\psi_m$ of 10,000 for the unreactive layer. Fluids

supplied to the base of the model domain percolate upwards through this layer and begin reacting with the reactive material above it. The scenario is akin to fluid migration away from a fluid-supplying channel, such as a pre-existing fracture, a fault, and a subduction interface, represented by the base of the model, into the neighboring rock.

Models were run to a bulk reaction completion of 1.5–3.0% depending on the computation time. Previous work has shown that

existing fractures accommodate subsequent deformation past a few reaction percent (*Yoshida et al., 2020; Shimizu and Okamoto, 2016*), and field observations indicate the majority of reaction-induced cracking occurs during early stages of serpentinization



(e.g., *Kelemen and Hirth, 2012*). Furthermore, at the conditions of the models, the volume expansion is so large that stresses sufficient to fracture the model can be achieved at only small reaction percentages. Therefore, the simulated degree of reaction completion is likely sufficient to capture the primary fracture behavior.


We limit our analysis to three stress conditions: hydrostatic, deviatoric tension, and deviatoric compression. We define the latter two conditions based on the orientation of the maximum compressive principal stress ($\sigma_1$) relative to the initial fluid-supplying channel (**Fig. 1**): $\sigma_1$ normal ($\sigma_1 = \sigma_y$) and parallel ($\sigma_1 = \sigma_x$) to the channel, respectively. The effective confining pressure in all simulations is 1 MPa unless stated otherwise. This relatively low confining pressure is used to ensure numerical stability in the

calculation of stress due to volume expansion as in previous studies. Given the sensitivity of reaction-induced fracture to numerous parameters, as established in previous studies (*Ulven et al., 2014b; Shimizu and Okamoto, 2016*), the results of this study are used to investigate the trend in the fracture behavior with differential stress, and the aim is not to investigate reaction-induced fracture in a specific tectonic environment. However, we later assess the implication of the modeling results to the interpretation of geophysical and geologic observations of hydration in the oceanic lithosphere at mid-ocean ridges and outer

rises and the forearc mantle wedge corner in subduction zones.

## 3 Results

We run simulations under deviatoric tension, a hydrostatic condition, and deviatoric compression (**Table S1; Fig. 2–5**). We first describe the model under a hydrostatic condition and use it as a reference model.

### 3.1 Reference Model under the Hydrostatic Condition

In simulations under the hydrostatic condition (1 MPa) (Model H1; **Table 2**; **Fig. 2c**), expansion of the bottommost reactive disks generates lateral compression amongst those reactive disks and lateral tension in the interior of the model, consistent with stress distributions observed in previous studies (*Ulven et al., 2014b*). The lateral tension in the interior causes localized sub-vertical mode I fractures to form. These tensile fractures are initially isolated from available fluids because the fluids cannot propagate across the layer of reacting disks that are under lateral compression. However, heterogeneities in the layer of reacting

disks and shear cracks that form with continued reaction facilitate further formation of tensile fractures across the layer of reacting disks and allow eventual connection between the interior tensile fractures and the fluid supply at the base of the model. These tensile fractures subsequently channel large volumes of fluid into the model. Fluids are consumed by disks along these newly fluid-filled channels, generating new tensile stresses parallel to the channels and causing a second generation of tensile fractures to form subnormal to the first generation of fractures. We categorize the fracture pattern that develops under the

hydrostatic condition as a branching fracture style. It is a result of hierarchical fracturing associated with a positive feedback loop, in which reaction-induced stresses generate tensile fractures that act as new fluid-supplying channels penetrating into the surrounding unreacted material, increasing its permeability and allowing further fluid infiltration and reaction. This branching process partitions the model domain into smaller subdomains that are bounded by newly generated fractures. This branching fracture is distinct from the branching fracture patterns that are described to occur during solid volume reducing reactions

(*Okamoto and Shimizu, 2015*). Because new tensile fractures propagate away from existing fluid filled channels, this fracture pattern consists of a relatively uniform mixture of low and high angle cracks (**Fig. 2c(ii)** and **3c**), and efficiently channels fluids into unreacted regions of the model, promoting spatially extensive hydration (**Fig. 4c(i–ii)**).

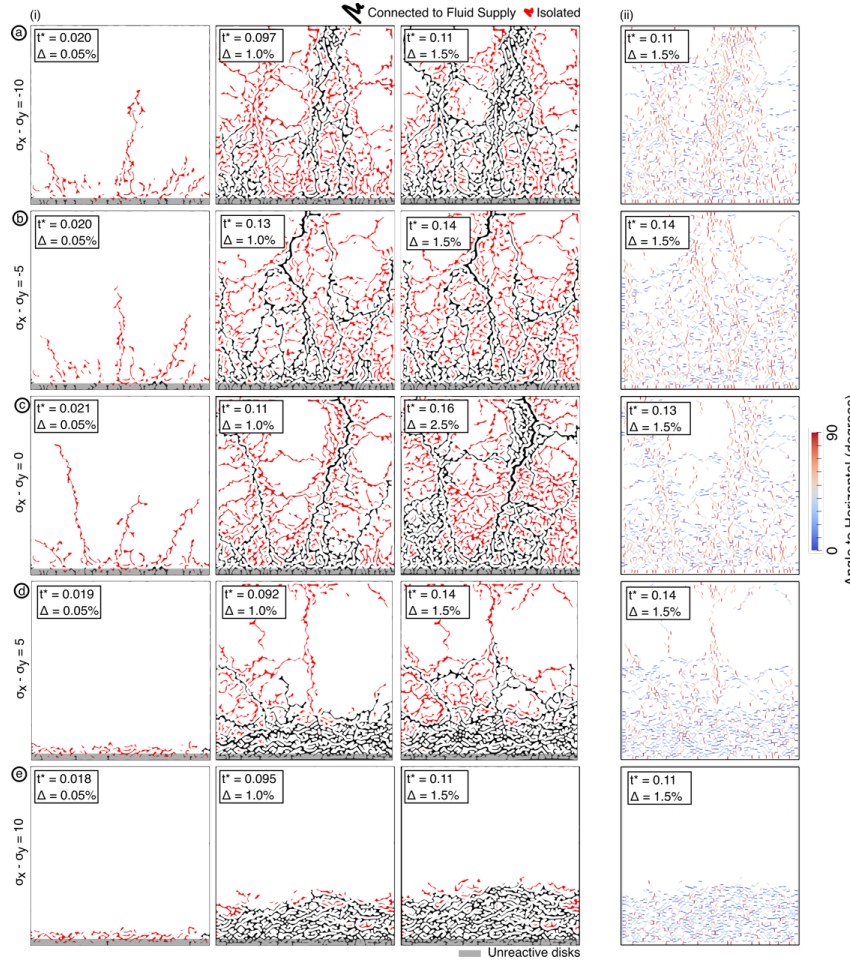


**Figure 2. (i) Fracture evolution (left three columns) and (ii) crack angle, ranging from 0º (horizontal; blue) to 90º (vertical; red) under (a, b) lateral deviatoric tension with differential stress (i.e., $\sigma_x - \sigma_y$) of –10 MPa and –5 MPa, respectively, (c) hydrostatic condition, and (d,e) lateral deviatoric compression with differential stress of 5 MPa and 10 MPa, respectively. t\* is the nondimensionalized time. Black and red paths indicate cracks that are connected to and isolated from the fluid supply at the base of the model, respectively. The**

**gray layer at the bottom of the domain represents a region with unreactive disks. Hydrostatic condition is shown up to $\Delta$ = 2.5% to better illustrate progressive hierarchical fracturing.**



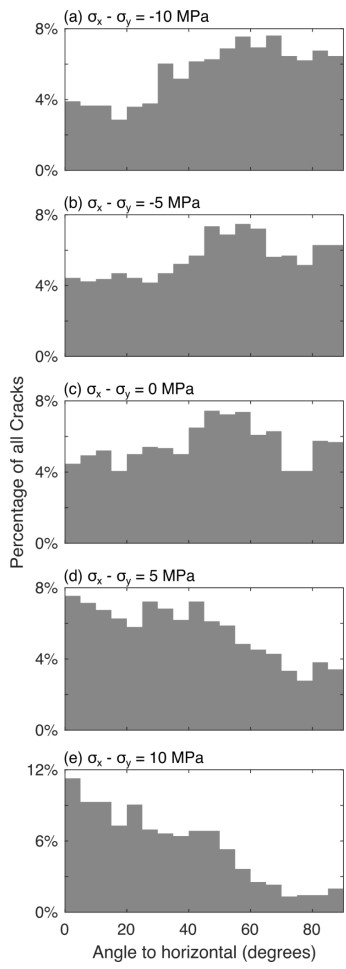

**Figure 3. Distribution of crack angles at $\Delta = 1.5\%$ under (a, b) lateral deviatoric tension with differential stress (i.e., $\sigma_x - \sigma_y$) of –10 MPa, –5 MPa, (c) hydrostatic condition, and (d, e) lateral deviatoric compression with differential stress of 5 MPa and 10 MPa, respectively.**




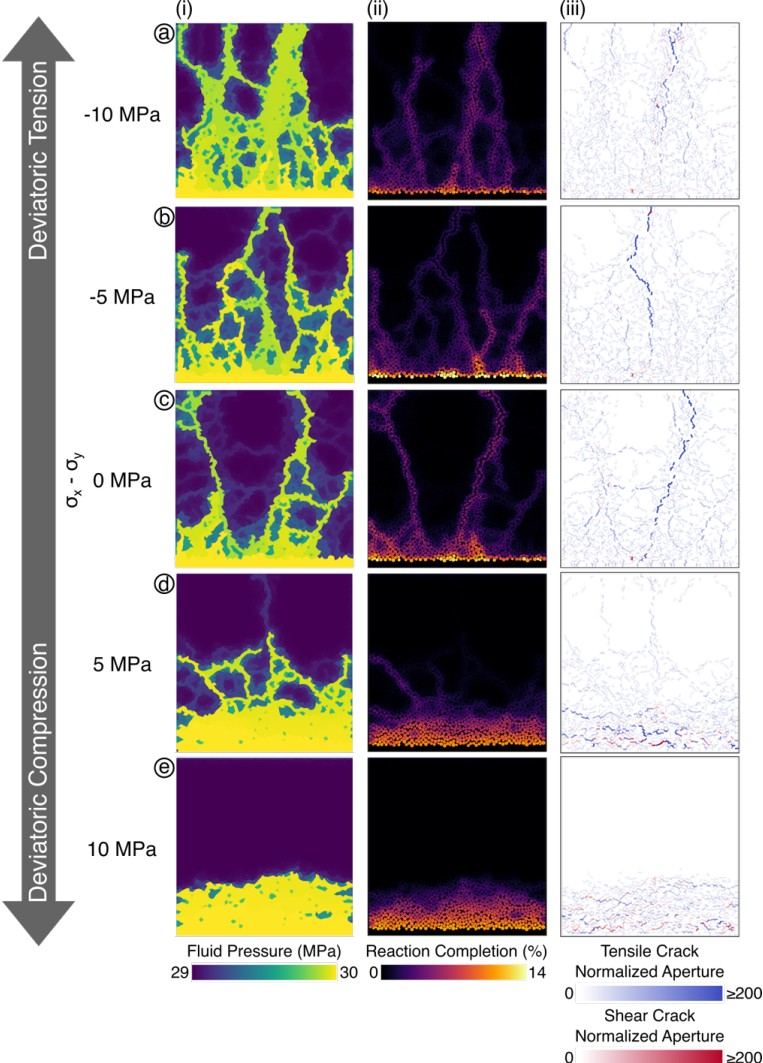

**Figure 4. (i) Fluid distribution, as measured by pore fluid pressure, (ii) reaction completion, indicating the level of volume expansion, and (iii) crack aperture at $\Delta = 1.5\%$ under (a–e) lateral deviatoric tension with differential stress (i.e., $\sigma_x - \sigma_y$) of –10 MPa, –5 MPa, hydrostatic condition, and lateral deviatoric compression with differential stress of 5 MPa and 10 MPa, respectively. The aperture of tensile and shear cracks (orange and purple lines, respectively) are normalized by the aperture of bonded, unstressed disks (w0) that characterize the matrix permeability.**

We use the pore fluid pressure distribution to assess fluid migration into the system through fractures (**Fig. 4c(i)**). High fluid pressures are localized along branching fractures that are connected to the fluid-supplying channel but decrease with distance from the fluid supply at the base of the model. The evolution of fluid pressure does not strictly follow that of fracture as many fractures are neither directly nor indirectly connected to the fluid supply at the base of the model. High fluid pressures are constrained to the cracks that are connected to the fluid supply because the matrix permeability is low and there is little fluid



migrating away from the cracks. The hydration distribution generally follows the fluid distribution with some delay due to the
time-dependence of hydration, and the highest reaction completion is along the base of the model (**Fig. 4c(ii)**). Both the fluid and
hydration distributions are impacted by crack aperture, which controls the fracture permeability. Fractures that are oriented
normal to the initial layer of reacting disks have the largest aperture and therefore host the highest fluid flow rates, playing a
critical role in hydrating the interior (**Fig. 4c(iii)**). As new layers of reacting disks develop at the boundaries of the first
generation high aperture cracks, the resulting tensile stresses will then increase the aperture of a second generation of high
aperture cracks normal to the first, allowing efficient flow into subdomains. This results in a pore pressure distribution
distributed throughout the interior of the model but localized along the fractures and a reaction distribution that is relatively low
but distributed throughout the model along fluid channels, where it is localized.

New fractures are not immediately connected to the fluid supply to form the next generation of fluid channels, and therefore the
overall fracture evolution exhibits a cyclic behavior, interceded by periods of stress buildup. Such cyclic behavior has been
observed and described in previous numerical modeling studies (*Okamoto and Shimizu, 2015; Shimizu and Okamoto, 2016*) and
experiments (*Okamoto et al., 2025*). These successive branching episodes result in a stepwise increase in fracture density (that is,
the percentage of connecting bonds that are broken in the model) and flow rate into the model (**Fig. 5a, b**), and they correspond
to sudden increases in the bulk reaction rate ($d\Delta/dt$) (**Fig. 5c**). Because the length of vertical tensile fractures is limited by the
top boundary of the model, fluid inflow is likely underestimated once the fractures reach the top boundary (**Sect. S3**). Therefore,
we refrain from directly comparing them with the inflow and reaction rates from other models.



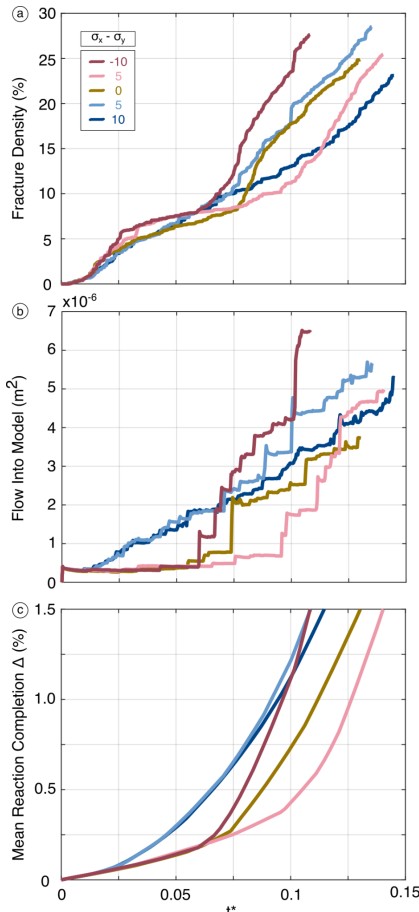

**Figure 5. Changes in (a) crack density at the base of the model, (b) fluid inflow rate, and (c) the bulk reaction completion (Δ) for the**
**entire model domain with nondimensional time. In the tension and hydrostatic cases, fracture density increases following each fluid in-**
**flow pulse, before beginning to level off. The relative magnitudes of the values in (a–c) depend on the model dimensions (Sect. S3; Fig.**
**S3). Therefore, they cannot be directly compared with those measured in laboratory experiments.**

## 3.2 Models under Horizontal Deviatoric Tension

The fracture evolution under vertical deviatoric tension is initially similar to those under the hydrostatic condition (Models T1
and T2; **Table 2**; **Fig. 2a(i),b(i)**). Vertical tensile cracks propagate from the base of the model into the unreacted interior.
Subsequent evolution indicates that when those fractures channel fluids, new cracks form normal to them, as in the hydrostatic
case, but also parallel to them (**Fig. 2a(ii),b(ii)**), resulting in more cracks with high angles relative to the distribution under the
hydrostatic condition (**Fig. 3a,b**). This style of fracturing appears to be a hybrid of both the branching fracture described in the
hydrostatic case, and a fracture style similar to "spalling" reported in previous studies (*Ulven et al., 2014b, Jamtveit et al., 2009;*
*Røyne et al., 2008*). The spalling cracks are characterized by short, distributed fractures between and very near to reacting disks
and, in our simulations, develop adjacent to fluid-filled channels that are parallel to $\sigma_1$. Spalling cracks generally appear as layers
of parallel cracks along fluid channels (**Fig. 2a(ii),b(ii)**). Note that in the hydrostatic model, while cracks around vertical



channels are largely of the branching type, some spalling cracks may develop along fluid channels due to the local deviatoric compression that develops due to the expansion of disks along the channel, but they are much less prevalent than in the models under tension.

Similar to the hydrostatic case, the distributions of fluid and hydration follow the distribution of fractures that are connected to the fluid supply (**Fig. 4a(i,ii),b(i,ii)**). However, due to spalling around fluid channeling fractures, high pore pressure occurs in a
wider zone along the channels. As a result, the width of reaction distribution along channels is large and increases as spalling occurs. Cracks oriented vertically have the highest apertures (**Fig. 4a,b(iii)**). However, apertures in models under deviatoric tension are generally smaller than under hydrostatic conditions, likely because the larger number of fractures around the main channels leads to reaction-induced stresses that discourage the widening of neighboring channels (i.e., larger $F$ in Eq. 2 in **Methods**). Both the fracture density and the inflow rate exhibit the same cyclic behavior as in the hydrostatic case, but due to the
large number of vertical fractures, the average increase in inflow rate is significantly higher than in the hydrostatic case (**Fig. 5**).

### 3.3 Models under Horizontal Deviatoric Compression

Fracture evolution under horizontal deviatoric compression ($\sigma_x - \sigma_y = 5$ and 10 MPa) initially exhibits behavior that is significantly different from that of the hydrostatic and deviatoric tension conditions (Models C1 and C2; **Table 2**; **Fig. 2d(i),e(i)**). Fracture evolution in these simulations occurs in two stages: an early stage dominated by spalling fracture, and a later
stage dominated by branching fracture. The timing of the transition from the early to the later stage depends on the magnitude of deviatoric compression.

In the early stages of hydration, expansion of the bottommost disks generates compression in the reacting region, similar to those under the hydrostatic condition and horizontal deviatoric tension. However, cracks develop between and near the reacting disks
before significant tensile stresses are able to build up in the unreacted interior. These cracks are both tensile and shear although tensile cracks are more common, and their pattern resembles the spalling fracture reported in the deviatoric tension simulations. They are predominantly at low angles and subparallel to the fluid supply at the base of the model, although there are a small number of high angle cracks (**Fig. 2d(ii),e(ii)**), resulting in more cracks with low angles relative to the distribution under the hydrostatic condition (**Fig. 3d,e**). The interior of the model remains under horizontal deviatoric compression due to background
stress, preventing the propagation of fractures into the unreacted interior.

Unlike in the hydrostatic and horizontal deviatoric tension cases, long sub-vertical tensile fractures do not form under horizontal deviatoric compression. Without those fractures, fluids are not channeled far into the unreacted rock, and spatially pervasive reaction does not occur throughout the model (**Fig. 4d(i,ii),e(i,ii)**). Instead, the spalling fracture pattern results in high fluid
pressures and reaction degrees in a wider zone along the fluid supply at the base of the model. This zone grows in width as spalling gradually propagates into the interior of the model. In these simulations, fractures with the largest aperture tend to be oriented subparallel to the fluid supply at the base of the model (i.e., in the direction of maximum principal stress) (**Fig. 4d(iii),e(iii)**). All cracks near the base of the model quickly become well connected to each other and to the fluid supply. As a result, fluid flow into the model correlates more closely to the crack density than in the hydrostatic or deviatoric tension
simulations (**Fig. 5**).



When the magnitude of horizontal deviatoric compression is small (i.e., $\sigma_x - \sigma_y = 5$ MPa), once the spalling region expands sufficiently far into the model, a second stage occurs wherein tensile cracks develop at high angles to the layer of spalling fractures. This occurs because the remaining unreacted part of the rock can no longer support the tension due to the expansion of

the reacted disks. For this reason, simulations in which the model domain is taller, the second stage occurs later (**Sect. S3**). The tensile cracks that form as part of the second stage propagate into the unreacted interior, and a branching fracture pattern develops in their vicinity. However, they occur at a shorter length scale and curve sharply to align themselves with the applied horizontal compression. This second stage of fracturing occurs earlier in the simulation with a lower degree of compression (i.e. $\sigma_x - \sigma_y = 5$ MPa) than in the more compressive case ($\sigma_x - \sigma_y = 10$ MPa), indicating that the timing of this second stage

depends on the magnitude of horizontal deviatoric compression.

## 4 Discussion

### 4.1 Primary Fracture Style and Its Transition with Differential Stress

Under the hydrostatic condition (Model H1; **Fig. 2c**), the reaction-induced local deviatoric tension that initially develops in the interior due to the hydration of the bottom most disks is parallel to the fluid-supplying channel, resulting in the branching

primary fracture style, as discussed in **Sect. 3.1**. Under background differential stress, the primary fracture style depends on the orientation of the fluid-supplying channel relative to $\sigma_1$. Under tension (Models T1 and T2), the initial fluid-supplying channel, corresponding to the bottom boundary of the model, is normal to $\sigma_1$, resulting in branching style fracture. Conversely, under compression (Models C1 and C2), the initial fluid-supplying channel is parallel to $\sigma_1$, resulting in spalling style fracture as the dominant style. As discussed in **Sect. 3.1 and 3.2**, some spalling does occur along subvertical fluid-supplying channels under

tension, because these channels are parallel to $\sigma_1$ in addition to the reaction-induced local deviatoric stress imposed by reacting regions, similar to the model under the hydrostatic condition. However, these spalling fractures are secondary.

Applied compression promotes spalling parallel to fluid channels by working with compressive reaction-induced stresses and working against tensile reaction-induced stresses. In our simulations, the transition in the primary fracture style occurs under

compression near 5 MPa (Model C1). Near this transition, spalling is still the primary fracture style, but it gives way to branching after sufficient reaction completion. While a transition from branching to spalling with increasing background compression parallel to fluid-filled channels is a general trend, the background stress value at the transition is specific to our model setup because it is sensitive to the reaction-induced stress distribution, which depends on factors that may vary between individual settings, such as the initial permeability, reaction-induced volume change, and domain size (*Ulven et al., 2014a,b;*

*Røyne et al., 2008;* **Sect. S4**).

The occurrence of branching after some amount of spalling near the transition (Model C1) is consistent with field observations of weathered rocks (*Røyne et al., 2008; Iyer et al., 2008*), with numerical models (*Røyne et al., 2008; Jamtveit et al., 2009*), and with observations from laboratory experiments (*Okamoto et al., 2025*). Previous authors have argued the transition to branching

to primarily be a result of changes in the unreacted domain size as spalling progresses (*Røyne et al., 2008*). This argument is consistent with our modeling results, which exhibit branching when spalling fractures have extended far enough into the model as discussed above. Those spalling fractures efficiently channel fluids into the rock in our models, and therefore the pore pressure and resultant reaction-induced stress distribution change as spalling progresses. The primary reaction-induced fracture



style under channel-parallel compression is then time-dependent, and spalling may represent only a first stage of reaction-
induced fracture. The orientations of branching fractures that occur under compression after spalling still follow the primary
orientations of the spalling fractures and tend to be parallel to the maximum applied background stress because they curve to
orient themselves with the background $\sigma_1$. At higher compression, this fracture pattern evolution can happen after a longer period
of simulation/hydration.

The transitional background stress of 5 MPa from our simulations reflects the magnitude of the local deviatoric stress that offsets
the background stress. Based on upper bound thermodynamic estimates of the crystallization stress due to serpentinization,
tensile crack propagation characteristic of the branching fracture pattern should be possible for background stresses of up to 100s
of MPa (*Malvoisin et al., 2017*). The apparent discrepancy in the background stresses for branching fracture between the
thermodynamic calculations and our models may occur because spalling cracks accommodate strain and hinder subsequent stress
build-up. *Kelemen and Hirth (2012)* noted a similar contrast between theoretical and observed reaction-induced stresses in
natural serpentinites and attributed the differences to strain accommodation by existing fractures. Although the exact value of
differential stress for the fracture style transition in our simulations is specific to the model parameters and the initial and
boundary conditions used, the results indicate that the style of reaction-induced fracture may be sensitive to a relatively small
change in the background stresses (e.g., a change from 0 to 10 MPa).

**4.2 Fluid and Hydration Distributions**

In both fracture styles, differential stress impacts the primary orientation of high-aperture cracks. Therefore, the contrasting
fracture styles produced under differing background stress conditions might have significant impacts on fluid flow during
hydration. Branching fracture results in a distributed fracture network that promotes a spatially extensive fluid and hydration
distribution at the model scale. During branching, fractures that form sub-normal to fluid-supplying channels tend to have the
highest apertures and propagate far into unreacted regions, channeling fluids into the interior of unreacted rock. In contrast,
during spalling fractures parallel to fluid filled channels have the highest apertures, and the fluid and hydration distributions are
localized along fluid channels. A similar spalling fracture pattern along the fluid supply is observed in experiments at high mean
normal stress (*Okamoto et al., 2025*). Given the relative orientation of the fluid channel and spalling fractures, the region with
relatively high fluid content and degree of hydration tends to form parallel to $\sigma_1$. Additionally, the highest aperture cracks are
parallel to $\sigma_1$, potentially guiding mesoscale fluid flow direction under tectonic stress.

**4.3 Implications of Fracture Styles for Geological Processes and Comparison to Observations**

At mid-ocean ridge and outer rises, hydration of the oceanic crust and mantle occurs and plays an important role in sequestering
water into the oceanic lithosphere. Sequestered fluids are released during subduction of the oceanic lithosphere, and some of the
released fluids are taken up by hydration of the surrounding material, such as the overriding mantle wedge. Based on our
modeling results, whether the hydration is localized or distributed in these environments could potentially depend on the
orientations of fluid-supplying channels relative to $\sigma_1$ (**Fig. 6**).

At mid-ocean ridges and outer rises, the oceanic plate experiences approximately surface parallel deviatoric tension due to
gravitational forces that pull the oceanic plate downslope, and bending, respectively (**Fig. 6a**). In both settings, the tectonic stress
state results in normal faulting that provides an initial fluid channel into the lithosphere (*Faccenda et al., 2009; Seno and*



*Yamanaka, 1996*). However, fluid migration away from these faults into the wall rock likely depends on the microscopic permeability of the wall rock (*Hatakeyama et al., 2017*), which may be impacted by reaction-induced fracture. The faults are subvertical and subparallel to $\sigma_1$, similar to the initial boundary conditions of models under compression (Models C1 and C2). These models predict spalling and localized hydration along the normal faults. Seismic and electromagnetic surveys indicate the

presence of deep cutting normal faults and hydration along them (*e.g., Ranero et al., 2003; Naif et al., 2015; Obana et al., 2019*). Although the length scale of our simulations is much shorter, the model-predicted spalling fracture style may contribute to the inferred localization of hydration along deep-cutting faults.

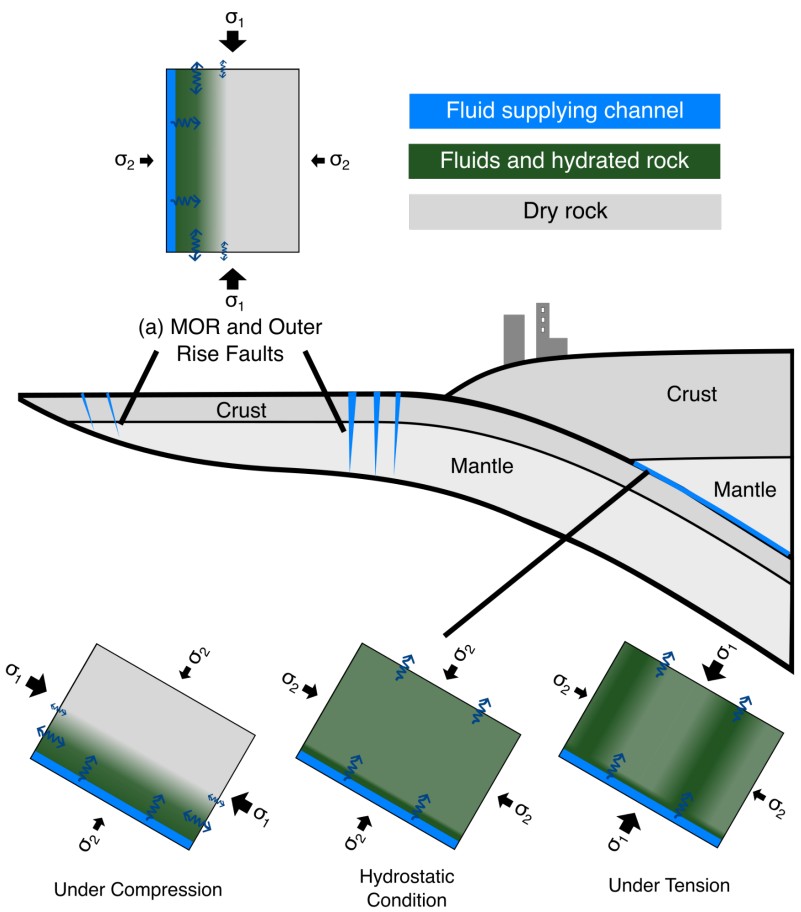

**Figure 6. Schematic of possible fluid and hydration distributions that might be expected from the evolution of fracture due to volume-increasing hydration at (a) mid-ocean ridges and outer rises and (b) the base of the mantle wedge corner under compression, hydrostatic condition, and tension.**

Hierarchical fracture networks are frequently reported in serpentinites from mid-ocean ridges (*Shimizu and Okamoto, 2016; Malvoisin et al., 2017; Roumejon and Cannat, 2014; Boudier et al., 2009; Jamtveit et al., 2009; Raleigh and Paterson, 1965*).



The hierarchical fracture network shares a similar fracture pattern to branching fracture in our hydrostatic simulations, but contrasts with the spalling that dominates under compression. Therefore, model-predicted spalling around the normal faults is inconsistent with textures of mid ocean ridge serpentines. Since the transition from branching to spalling is predicted to occur

under some compression parallel to the fluid channel, this may imply that the vertical stress relative to the horizontal stress may be too low to promote spalling. However, simulations under low differential stress (C1) indicate that the fracture pattern changes from spalling to branching after the spalling front has migrated a sufficient distance into the model, and the observed hierarchical fracture may indicate the fracture pattern in the later stage of hydration. Another interpretation is that the samples exhibiting hierarchical fracture may represent reaction-induced fracture along fluid-supplying channels that form sub-normal to the normal

faults. Alternatively, high initial permeability may promote branching, per previous studies (*Ulven et al., 2014b*).

At subduction zones, geophysically-inferred stress orientations and lithospheric deformation models indicate that the state of stress in the overriding lithosphere in the forearc region can be horizontal margin-normal deviatoric compression to tension (**Fig. 6b**) and has been attributed to the variation of the plate coupling force relative to the gravitational force (*Wang and He, 1999*;

*Sharma et al., 2025*). The overriding forearc mantle wedge corner in subduction zones is decoupled from the subducting slab and cold (e.g., *Furukawa, 1993; Wada et al., 2009*), acting as part of the overriding lithosphere. The distribution of hydration by slab-derived fluids may be impacted by reaction-induced fracture under background tectonic stress. Assuming that the subduction interface represents a fluid-supplying channel and that the interface is shallowly dipping, the horizontal margin-normal deviatoric compression and tension correspond to channel-parallel compression and tension for the initial stage of

hydration, respectively. Our models indicate that these stress conditions promote spalling and branching, respectively.

In forearcs under margin-normal horizontal deviatoric tension and a near hydrostatic condition, hydration might instead be accompanied by branching fracture or a hybrid of branching and spalling similar to those predicted by Models H1, T1, and T2. Fractures in this case efficiently channel fluids into the interior of the mantle wedge corner and facilitate extensive hydration. In

fact, In forearcs under margin-normal deviatoric tension or where the margin-normal and vertical stresses are similar, such as SW Japan (*Saito et al., 2018*) and Cascadia (*Wang and He, 1999; Balfour et al., 2011; Sharma et al., 2025*), a relatively high degree of mantle wedge corner serpentinization is reported (*Bostock et al., 2002; Saita et al., 2015; Kato et al., 2010; Kamiya and Kobayashi 2000*). The branching fracture pattern is similar to hierarchical fracture patterns exhibited in exhumed serpentinites and carbonates from, for example, the New Caledonia and western Mongolia paleo-subduction zones (*Dandar et*

*al., 2019; Raia et al., 2022*), potentially providing clues to the background tectonic stresses.

In forearcs under margin-normal horizontal deviatoric compression, more localized hydration at the base of the mantle wedge corner is expected due to the model-predicted spalling style fracture. Such spalling fracture might guide fluids updip along the interface and prevent extensive fluid flow into the mantle wedge corner. In fact, there are little observations that indicate

spatially extensive serpentinization in the mantle wedge corner under margin-normal compression, such as in NE Japan (*Miura et al., 2005*) and Chile (*Carlson and Miller, 2003*). Furthermore, localized hydration at the base of the mantle wedge has been invoked to explain high Poisson's ratios observed atop the subduction interface in much of NE Japan (*Kawakatsu and Watada, 2007*). Experimental results report localized fluid flow within sheared serpentinites due to the resulting permeability anisotropy (*Kawano et al., 2011; Okazaki et al., 2013*). This mechanism may be operating in parallel with the mechanism we propose here

for updip fluid migration. Exhumed mantle wedge serpentinites from the Sanbagawa belt exhibit foliations containing antigorite



with textures that are somewhat comparable to the spalling in our horizontal compression models (*Mizukami et al., 2014; Okamoto et al, 2024*).

### 4.4 Limitations of This Study

In this study, simulations were run at low effective confining pressure of 1 MPa, which may be lower than the effective
confining pressure in the tectonic environments discussed above. Hydration may occur at overburden pressures of up to several
100 MPa along normal faults in mid-ocean ridges and outer rises and around 1–2 GPa at the base of the mantle wedge corner.
Although the effective confining pressure is likely to be lower due to pore fluid pressure, further investigation on the role of
confining pressure is required to extend quantitative analyses of reaction-induced fracture to these environments.

The primary fracture style is also dependent on the domain size relative to the thickness of the reacting layer (*Royne et al., 2008*),
which in turn depends on $\psi_m$. Previous numerical modeling results and field observations indicate that larger domains or
increasing the model height promote spalling (*Ulven et al., 2014a; Jamtveit et al., 2009*; **Sect. S2**). Decreasing the matrix
permeability, increasing the porosity, or adjusting the volume of fluids consumed by the reaction also promote spalling through
their impacts on $\psi_m$ (*Ulven et al., 2014b;* **Sect. S4**). However, models with these parameters that encourage spalling eventually
transition to branching after the layer of spalling advances to sufficient thickness. Therefore, the transition in the fracture style is
dependent on the duration of hydration in addition to the aforementioned model parameters although this does not impact the
trend in the fracture style with the background stress.

### 5 Conclusion

Distinct element simulations indicate that reaction-induced fracture patterns are sensitive to background tectonic stresses.
Background stresses that are hydrostatic or have $\sigma_1$ perpendicular to fluid-supplying channels promote extensive hydration and
branching fracture, whereas background stresses parallel to fluid-supplying channels promote spalling and localization of
fracture and hydration along the channels. Spalling in the channel-parallel compression case is sensitive to the magnitude of
deviatoric compression, as well as the size of neighboring unreacted domains. As spalling propagates into unreactive rock, the
fracture style may switch to branching when the remaining unreacted domain becomes small. Based on the modeling results,
hydration likely localizes along vertical faults at mid-ocean ridges and bending faults in the outer-rise, given $\sigma_1$ is vertical and
sub-parallel to those faults. At the base of the mantle wedge corner, hydration may localize along the subduction interface if $\sigma_1$ is
margin-normal or be spatially extensive if $\sigma_1$ is vertical.

### Data Availability

The numerical modeling results are available through Zenodo at 10.5281/zenodo.17121292.

### Code Availability

The DEM code used in this study is detailed in *Shimizu et al. (2011)* and *Okamoto and Shimizu (2015)* and is available upon
request.



**Author Contributions**

**J.J. McElwee:** Conceptualization, Investigation, Modeling, Interpretation, Writing, Visualization, Project administration. **I.**
**Wada:** Conceptualization, Interpretation, Writing, Supervision, Project administration, Funding acquisition. **H. Shimizu:** Code
development. **K. Yoshida:** Conceptualization, Interpretation, Writing. **A. Okamoto:** Conceptualization, Interpretation, Writing.

**Competing Interests**

The authors declare no competing interests.

**Acknowledgements**

This work is supported by NSF Grant EAR-1847612 to IW.

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
