# Peer review of "Impact of differential stress on fracture due to volume increasing hydration"

_EGUsphere, 2025_

## Author Comment (AC1)

**Response to Review #1**

We thank the reviewer for taking the time to read the manuscript and for providing feedback. We have addressed their comments below. The reviewer's comments are in plain text. Our response is in blue, and the excerpts of revised text in the manuscript relevant to the comment are indented, with line numbers provided. Please note that when we refer to a figure, we use the figure number from the revised manuscript.

**Review 1:**

Through observations in natural rocks, experiments and previous models it is well known that volume increasing hydration reactions, such as serpentinization, lead to fracture nucleation, i.e., reaction-induced fracturing. In their manuscript McElwee et al. bring this process a step forward by investigating how tectonic stresses in various settings influence fracture propagation. Through numerical models they test different stress configurations and find that large fracture networks branching into the surrounding rock form in tensile regimes. To the contrary, in compressional regimes such networks do not form, or only when the reaction is already well advanced, with sever implications on the hydration stage of mid ocean ridges and bending faults. These results are significant and certainly of interest for the community. I only have a few minor comments.

The manuscript is well written and I really enjoyed reading it. Specifically, I acknowledge the detailed discussion on model limitations. All models were run at 1 MPa confining pressure while it is known from experiments that high confining pressures inhibit fracture nucleation. However, I miss a similar discussion on the effect of temperature. We know that the serpentinization rate is sensitive to temperature and maximum reaction rates are reached at 270 – 300 °C. Within the mantle wedge we expect strong temperature gradients, such that reaction rate varies in space as do elastic parameters.

Reaction rate variation that arises due to temperature variation impacts the reaction-induced strain gradient and thus reaction-induced fracture (*Dargi et al., 2025; Shimizu and Okamoto, 2016*). However, because of the low permeability of mantle rock (*Katayama et al., 2020*), the fluid pressure gradient in the region of our interest (where serpentinization is occuring) is likely much larger than the temperature gradient. Additionally, the temperature gradient can probably be incorporated into the Damkholer number, which has previously been taken to define the sharpness of reaction-induced strain gradients (*Shimizu and Okamoto, 2016*).

In other words, when the reaction is fastest the mechanical behavior may favour visco-elasto-plastic rather than brittle responses to the reaction. At higher temperature, the reaction rate slows down, further supporting non-brittle behavior due to decreased strain rates.

We agree with the reviewer that the pattern of reaction-induced fracture may be sensitive to temperature because temperature impacts the reaction rate and thus strain rate. We have some ongoing work on this topic. For now, we have added the following to the manuscript (**Line 825**):

> Additionally, serpentinization and therefore reaction-induced fracture may occur at temperatures up to ~600C in the mantle wedge corner, with the warmest mantle wedge corners, such as in Cascadia, being between 400 and 600C in their entirety (*Wada and Wang, 2009*). At these temperatures, the serpentinization rate is much lower than its maximum rate at roughly 300C (*Malvoisin et al., 2012*), and, therefore, the strain rate due to the volume increasing reaction is probably low. The plastic yield stress of serpentine depends positively on the temperature and negatively on the strain rate (*Burdette and Hirth, 2022; Horn and Skemer, 2025*), and so at these temperatures, non-dilatant plastic deformation may occur (*Malvoisin et al., 2021; Skarbek et al., 2018*) if the plastic yield stress is lower than the brittle strength. This style of deformation, which would likely accommodate strain but does not generate fluid pathways, is not included in our models. The brittle strength is proportional to the effective confining pressure, and so non-dilatant plastic deformation may not be present when the pore pressure is high.

While certain bonds will break and form new fluid pathways, others will ultimately close, which is the often discussed processes of clogging. How exactly is this treated in the model?

We have added the following to the manuscript (**Line 834**):

> Clogging due to precipitation of reaction products in fluid channels is not considered in the models presented here. The entirety of the volume increase goes into elastic and brittle deformation, and the bulk crack density of the model can only increase as the simulation progresses. However, crack and matrix permeability may locally decrease due to elastic stresses that narrows fluid channels at cracks and bonds, as indicated in Figure 4, which may limit supply of fluids to branching cracks. This scenario may be representative of natural serpentinization, given the low permeability of the lithosphere (*Katayama et al., 2020*), which favors fracturing (*Uno et al., 2022*), and the abundance of reaction-induced fracture textures in natural rocks (*O'Hanley, 1995; MacDonald and Fyfe, 1975*).

Furthermore, the volume change may be slightly dependent on pressure and temperature. Possibly this goes too far for this manuscript, but it might be interesting to test how temperature and pressure will affect the volume change and thus the fracture propagation in various tectonic settings.

We are not sure if the reviewer is referring to the solid or total volume change. The dependence of the solid volume change on pressure and temperature is likely small compared to its dependence on other factors, such as mass transport. However, it has been shown that the solid

volume change relative to the Damkholer number does indeed impact the fracture behavior (*Ulven et al., 2014b*). The general impact of the solid volume change is noted on **line 344**. Furthermore, the fluid volume change is also sensitive to temperature and pressure. Its effect can potentially be implicitly tested through the Damkhoer number (*Ulven et al., 2014b*). However, the detailed analyses of the effect of solid and fluid volume changes with temperature and pressure is beyond the scope of the current manuscript.

Line 10 (and throughout the manuscript): to refer to the process, change "reaction-induced fracture" to "reaction-induced fraturing".

We have implemented this change.

Line 40: It could be helpful for the reader to have a reference to figure 6 here.

We have moved Figure 6 to Figure 1 and added a reference at **Line 46**.

Figure 6: In this figure, the compressional and extensional regimes within the mantle wedge could be labelled/highlighted in order to help the reader.

We are not sure if the reviewer means to label the model cartoons, or the mantle wedge corner itself. There are currently labels for the model cartoons. For the mantle wedge corner itself, the stress state likely varies among different subduction zones, and there may be even local variations along strike or dip. See the discussion on **Lines 532–535 and 548–550**.

Line 110: How are the values of P$min$ and P$max$ determined?

The relatively low Pmax and Pmin values of 29 and 30 MPa, respectively, are chosen such that pore fluid pressure does not cause hydrofracture in the model, as discussed on **Line 127**, but the exact values are otherwise chosen to be consistent with previous work (*Okamoto and Shimizu, 2015; Shimizu and Okamoto, 2016*). We also add some additional reasoning to the text (**Line 312**):

> Additionally, the difference between $P_{max}$ and $P_{min}$ is much smaller than the pressure decrease caused by 100% reaction of an average radius disk. As a result, the reaction is limited by the continual supply of fluids to pore spaces.

**References from response text that is not part of a manuscript addition:**

Dargi, M. A., Detournay, E., and Le, J.-L.: Eigenstrain-Induced Stress in Elastic Cylinder, Journal of Applied Mechanics, 93, 011006, https://doi.org/10.1115/1.4070149, 2026.

Katayama, I., Abe, N., Hatakeyama, K., Akamatsu, Y., Okazaki, K., Ulven, O. I., Hong, G., Zhu, W., Cordonnier, B., Michibayashi, K., Godard, M., Kelemen, P., and the Oman Drilling

Project Phase 2 Science Party: Permeability Profiles Across the Crust‑Mantle Sections in the Oman Drilling Project Inferred From Dry and Wet Resistivity Data, JGR Solid Earth, 125, e2019JB018698, https://doi.org/10.1029/2019JB018698, 2020.

Okamoto, A. and Shimizu, H.: Contrasting fracture patterns induced by volume-increasing and -decreasing reactions: Implications for the progress of metamorphic reactions, Earth and Planetary Science Letters, 417, 9–18, https://doi.org/10.1016/j.epsl.2015.02.015, 2015.

Shimizu, H. and Okamoto, A.: The roles of fluid transport and surface reaction in reaction-induced fracturing, with implications for the development of mesh textures in serpentinites, Contributions to Mineralogy and Petrology, 171, 1–18, https://doi.org/10.1007/s00410-016-1288-y, 2016.

Ulven, O. I., Storheim, H., Austrheim, H., and Malthe-Sørenssen, A.: Fracture initiation during volume increasing reactions in rocks and applications for CO2 sequestration, Earth and Planetary Science Letters, 389, 132–142, https://doi.org/10.1016/j.epsl.2013.12.039, 2014.

---

## Author Comment (AC2)

**Response to Review #2**

We thank the reviewer for taking the time to read the manuscript and provide feedback. The clarity of the manuscript has been improved by incorporating the reviewer's suggestion. The reviewer's comments are in plain text. Our response is in blue, and modifications to the text are indented, with line numbers provided. Please note that when we refer to a figure, we use the figure number from the revised manuscript.

**Review 2:**

This manuscript explores the effect of differential stress on reaction hydration dynamics. The results show that the patterns of fractures and fluid pathways depends on the differential stress: 1) in tension (sigma1 < sigma 2 where sigma1 is normal to the side boundaries of the model domain in a pure shear sense), the branching fracture patterns are dominant, which helps with fluid reactions in the interior of the domain, thus further fracturing the interior, and 2) in compression, (sigma1 > sigma 2), spalling patterns are dominant, which limits fluid reactions from happening in the interior of the domain that can further fracture the domain. The paper discusses the implications for a few relevant tectonic settings. Overall, the paper is very good and would be a useful contribution with minor revisions.

Major comments (and questions):

1. Methodology: It would be useful for readers to have a system of equations being solved here so that it is clear how the equations presented in the main text fit into the larger system of equations. The method can be clearer, i.e. the circular disks seem to not overlap but it would be helpful to explicitly state this. What are the boundary effects if there is no layer of unreactive disks and how was the thickness of this layer chosen? Why are the three other boundaries impermeable? It is unclear what the imaginary pipes between disks are (each disk pair has a pipe between them? how is the length of the pipe chosen? is this the full length of the two disk in the direction normal to both disks or perhaps just the overlap?) and are these already scaled to be 1% of the actual pore space (is the aperture is already 1%)? How are these imaginary pipes connected or how do they become connected? I assume these pipes are constant in aperture through its length. How do the water filled cracks become connected? When there is an isolated crack, does it ever increase in volume more than the amount of water that is already there since no fluid is coming in to fill in?

To address all the questions in this comment, we have expanded the methods of the manuscript, explicitly describing the methodology of *Okamoto and Shimizu, 2015* and *Shimizu and Okamoto, 2016,* and have added a cartoon of the pore network setup to **Figure 2**. However, for clarity we directly answer specific questions here as well:

It would be useful for readers to have a system of equations being solved here so that it is clear how the equations presented in the main text fit into the larger system of equations.

We have added equations describing the contact laws and explicit integration of particle displacement to the manuscript in **Lines 100–155**, as well as equations describing the fluid flow **(Lines 237–249)**.

"the circular disks seem to not overlap but it would be helpful to explicitly state this."

Overlap of disks is used to calculate the contact force and displacement; the remaining overlap upon disk displacement represents the elastic strain on the bond. However, the degree of overlap is small because the stiffness of the disk contacts is high. We add to **Line 141:**

> Compression results when disks overlap and tension results when bonded disks are displaced further from each other.

"What are the boundary effects if there is no layer of unreactive disks and how was the thickness of this layer chosen?"

We add the following text to the manuscript, addressing the question (**Line 377**):

> "…to minimize the effects of forced alignment of disks along the boundary (**Fig. 1**). Exclusion of this layer results in artificially high apparent strengths of the bonds between the first disks to react."

The boundary layer was chosen to be just thick enough that there is some irregularity at the initial reaction front, but thin enough so that the unreactive layer does not impact the bulk strength of the model.

"Why are the three other boundaries impermeable?"

The sides are impermeable (free fluid boundaries) to mimic standard experimental setup. When the top boundary is permeable, large flow rates develop across the model that significantly increase the computation time, and therefore we made it impermeable for computational efficiency. We add the following to the text (**Line 253**):

> …mimicking standard experimental setup (i.e. *Uno et al., 2022*), and they also prevent long computation times that arise from continuous, rapid flow within fractures between the fluid supply and draining boundaries

It is unclear what the imaginary pipes between disks are (each disk pair has a pipe between them? how is the length of the pipe chosen? is this the full length of the two disk in the direction normal to both disks or perhaps just the overlap?) and are these already scaled to be 1% of the actual pore space (is the aperture is already 1%)?

The flow algorithm follows the pore network method (*Fatt, 1956*). As written in **Lines 138–140**, "In both cases [bonded disk contacts and cracks], imaginary pipes are placed between disks to represent flow channels, and the volumetric flow ($Q$) through the channel is calculated using the laminar flow equation." These pipes do not store fluids. The length of the pipe is the geometric

mean of the radii of the disks defining the contact. The unstressed apertures of the pipes are not explicitly related to the porosity of the model, but are chosen relative to the reaction rate based on the desired inverse Damkohler number (phi_m). Therefore, while the pore spaces are scaled by 1%, the pipes are not. We have added additional discussion of the fluid flow algorithm to **Lines 235–249**, and added a cartoon to **Figure 2**.

How are these imaginary pipes connected or how do they become connected?

Pipes channel fluids between pores. Therefore, they are connected if they share a pore.

I assume these pipes are constant in aperture through its length.

The pipes are constant in aperture.

How do the water filled cracks become connected?

We clarify that all pores are water filled, but it is their fluid pressure that changes throughout the simulation. They become connected when they are part of a chain of pipes and pores that connect to the pores at the base of the model where the pore pressure is prescribed as a boundary condition. Note that in **Figure 3** we called the red and black features "cracks." We have changed this to "fractures" because these features are chains of cracks and pores, and apologize for confusion that this could have caused.

When there is an isolated crack, does it ever increase in volume more than the amount of water that is already there since no fluid is coming in to fill in?

The volume of pores and the aperture of cracks can all change due to elastic deformation. When the volume of a pore increases, and there is insufficient fluid supply to compensate, the fluid pressure decreases. We have added **Eq. 7** and **Eq. 8**, which shows this, to the text.

2. Methodology: It is unclear how the volume increase is taken into account for the chemical reactions. Is it an isotropic volume increase (the disk increases in sizes and can overlap)? These models are in two dimensions, are there expected differences if you were to increase to a third dimension (spheres instead of disk)? How does the total solid + fluid volume imposed? Is it some local volume defined by a region surrounding the reacting disk? It is hard to understand how mass is conserved with all these parameterizations.

As with comment 1, we expand the methods section significantly. This should answer most of the questions in comment 2. However, for clarity we directly answer specific questions here as well:

It is unclear how the volume increase is taken into account for the chemical reactions. Is it an isotropic volume increase (the disk increases in sizes and can overlap)?

The volume increase is isotropic across each disk. The disk radius increases. We have added **Eq. 10** to the manuscript.

These models are in two dimensions, are there expected differences if you were to increase to a third dimension (spheres instead of disk)?

One possible impact is on the connectivity of fracture networks, which should be sensitive to the number of dimensions. However, this is beyond the scope of the current study.

How does the total solid + fluid volume imposed? Is it some local volume defined by a region surrounding the reacting disk? It is hard to understand how mass is conserved with all these parameterizations.

Fluids are consumed from pores. The solid volume increase upon 100% hydration is 50%, and the fluid volume decrease (consumption) is 70%. The net volume change of –20%.

The solid volume increases uniformly over a disk once it acquires a reacting surface, but the fluid volume decreases within pore space in a given hydraulic sub-domain based on the reaction rate, $Z$, which is uniform for all disks that sound the pore space, and the total volume of the surrounding disks within the hydraulic sub-domain (**Figure 2c**). We have added discussion to **Lines 328 – 333.**

The disks are randomly placed initially. If these models were to be repeated 1000 times, is there a spread of the model results? In particular, would you get the same answers from every model as shown in Figure 5?

Studies indicate that the variance in model behavior in uniaxial compression, uniaxial tension, and brazil tests depends on the disk count. We use ~3,150 disks, which is intermediate compared to similar studies that use around 400–10,000 disks. Based on our disk count, we expect a 7–15% coefficient of variation in the uniaxial tensile strength if we run strength tests on many models. While this could result in small changes to fracture patterns and to the stress at which distinct shifts between spalling and branching fracture occur, the overall trend should not be affected. We have included an additional set of simulations with different random initial disk configurations in a new section of the supplement (**Fig S2**) that show similar behaviors to those in **Figure 3–6**, and have added the following text to the main text (**Line 208**):

> Although the packing of disks in the model domain is random, previous work indicates that the variance of model behavior due to differences in the initial disk packing is small (*Shimizu et al., 2010*). An additional set of simulations with a different initial disk packing (**Fig. S2**) displays similar behavior to those in **Fig. 3–6**.

3. Why is it that the authors chose to run simulations at 1MPa which is not reflective of the environments that they are trying to capture? Why not run simulations at larger confining pressures?

Stability issues arise in the current formulation of the model at higher effective confining pressures (**Line 393**). Current work is in progress to investigate the impact of the effective confining pressure on reaction-induced fracture. The effective confining pressure in the mantle wedge corner is not well-constrained because the fluid pressure might vary from low to

lithostatic. Keeping the effective confining pressure constant in this study allows us to isolate the impact of differential stress. 1D stress profiles (*e.g. Ulven et al., 2014; Dargi et al., 2025*) seem to imply that the trend in the fracture evolution with differential stress identified in this study is likely to still hold at elevated effective confining pressure although the transition from branching to spalling patterns likely occurs at higher differential stress than predicted at 1 MPa effective confining pressure.

Minor comments

1. The figures are good at illustrating what is written in the main text but they can be improved. The figures themselves could be larger along with the labels. It would be clearer if there are labels for the models, i.e. T1,H1,C1 etc or tension/hydrostatic/compression like in figure 4. Typo in Figure 5 legend: there is a missing minus sign for the 5.

Thank you for pointing out the typo. We have added run labels to **Figures 3–6**. We have also increased the size of the figures which results in an increase in label fonts..

2. Line 159: How much faster are reaction rates here compared to nature? References for this? The ratios of fluid flow rates to reaction rates are a nice way to explore this but it is unclear how they fit into the system of equations and how realistic they are as compared to nature.

The rate of serpentinization observed in the lab depends on the temperature and grain scale (*Malvoisin et al., 2012*). However, the reported rate for 50–63μm diameter powders at 300C is equivalent to $Z_{max} \approx 10^{-8} s^{-1}$ in our model (*Shimizu and Okamoto, 2016; Malvoisin et al., 2012*). We add the following to the methods section **(Line 361)**:

> The chosen $Z_{max}$ is roughly 8 orders of magnitude larger than observed in lab experiments on olivine powders at 300C (*Shimizu and Okamoto, 2016; Malvoisin et al., 2012*).

3. What is Delta? This is not explained anywhere until stated as `bulk reaction completion' in Figure 5 caption. This needs to be explained much earlier since it appears in Figure 2,3,4. It is also confusing that there is a Delta in the supplementary material that is `average reaction degree' in section S2 and Figure S2. Please fix this.

We apologize for the error in the supplement. These Deltas represent the same quantity. Delta has been changed to bulk reaction completion in section S3 and **Figure S3**. We add a definition for delta to **Line 383**:

> Models were run to a bulk reaction completion, Delta, of 1.5–3.0% depending on the computation time.

4. A recent paper, Olive et al 2025, showed possible compression at the mid-ocean ridges that would be interesting to discuss.

This paper is relevant, especially considering the wide occurrence of mesh textures at mid-ocean ridges. We add the following discussion to **Line 674**:

Recent studies have reported that normal faults that form in the mid-ocean ridge environment may be reactivated as thrust faults as the oceanic plate moves away from the ridge (*Olive et al., 2024*), indicating that $\sigma_1$ can become subnormal to faults. Therefore, one additional possibility is that mesh-texture serpentinites formed under this condition, although it is not clear how much fluid can migrate down such faults that are being compressed.

5. It might be useful to the community to have a discussion in relation to what the results mean in the context of the fluid flow modeling at subduction zone that one of the authors is part of [Wilson et al 2014, Cerpa et al 2017,2019, 2025].

We have added the paragraph below, discussing the implication of our modeling results to fluid flow modeling results listed above by the reviewer (**Line 805**):

The results here are applied to the stagnant mantle wedge corner, which is at relatively low temperatures and likely undergoes brittle deformation. Modeling studies by Wilson et al. (2014) and Cerpa et al. (2017, 2019, 2025)on fluid flow in a viscously deforming solid, indicate compaction pressure gradients generated by deformation of the solid matrix may impact fluid migration (McKenzie, 1984; Spiegelman, 1993). While they apply the model to regions in subduction zones that are generally at higher temperatures and pressures (higher Maxwell number) where brittle deformation is discouraged, as discussed in Cerpa et al. (2025), brittle deformation, such as reaction-induced fracturing, may occur within their model domain of the subducting slab where the temperature is relatively low. Within the slab, both dehydration and hydration reactions can occur, potentially inducing reaction-induced fracturing. As shown in our study, the pattern of such fractures depends on the state of stress, impacting fluid migration. The slab tends to experience downdip tension at intermediate depths due likely to the slab pull, a key driving force of subduction (e.g., Isacks and Molnar, 1971; Vassiliou and Hager, 1988), but the details of stress distribution are impacted by various factors, such as the elastic bending of the slab at the trench and subsequent unbending downdip (e.g., Hasegawa et al., 1987), lateral bending due to margin curvature (Wada et al., 2010), and interaction of the slab with the transition zone and the lower mantle (Goes et al., 2017), and the age and the strength of the slab (Chen et al., 2004). Therefore, the style of reaction-induced fracture is likely to spatially vary with the stress orientations within the slab, and the combined effects of the permeability structure due to reaction-induced fractures and compaction pressure gradients due to viscous deformation may generate a complex pattern of fluid migration.

Synthesis of the two approaches would be valuable but is beyond the scope of the current modeling study.

6. In the supplementary section 1, when the authors talked about fully reacted case, do you mean to 3\% completion or everything is reacted?

These models are 100% reacted because their purpose is to calibrate the properties of 100% reacted disks. We have adjusted the wording from fully reacted to 100% reacted, and unreacted to 0% reacted.

7. Do the horizontal dimensions change the model results like the vertical extent does?

Because the fluid supply is roughly symmetric along the horizontal dimension, the horizontal dimension affects the decay of stress away from the reaction front (Saint Venant's Principle; *Timoshenko and Goodier, 1970*). Therefore, the effect of the horizontal dimension should become significant only when it is small. In the endmember where the horizontal dimension is very small, the tensile stress due to the volume expansion would be large and concentrated near the reaction front (e.g. *Dargi et al., 2025*), making branching fracture easier. As the horizontal dimension increases, the tensile stress depends on the height of the model as discussed in **Text S4**.

**References from response text that is not part of a manuscript addition:**

Dargi, M. A., Detournay, E., and Le, J.-L.: Eigenstrain-Induced Stress in Elastic Cylinder, Journal of Applied Mechanics, 93, 011006, https://doi.org/10.1115/1.4070149, 2026.

Fatt, I.: The Network Model of Porous Media, Transactions of the AIME, 207, 144–181, https://doi.org/10.2118/574-G, 1956.

Malvoisin, B., Brunet, F., Carlut, J., Rouméjon, S., and Cannat, M.: Serpentinization of oceanic peridotites: 2. Kinetics and processes of San Carlos olivine hydrothermal alteration, Journal of Geophysical Research: Solid Earth, 117, https://doi.org/10.1029/2011JB008842, 2012.

Okamoto, A. and Shimizu, H.: Contrasting fracture patterns induced by volume-increasing and -decreasing reactions: Implications for the progress of metamorphic reactions, Earth and Planetary Science Letters, 417, 9–18, https://doi.org/10.1016/j.epsl.2015.02.015, 2015.

Shimizu, H. and Okamoto, A.: The roles of fluid transport and surface reaction in reaction-induced fracturing, with implications for the development of mesh textures in serpentinites, Contributions to Mineralogy and Petrology, 171, 1–18, https://doi.org/10.1007/s00410-016-1288-y, 2016.

Timoshenko, S. P. and Goodier, J. N.: Theory of Elasticity, 3rd ed., McGraw-Hill Book Company, 1970.